# A New High-Throughput Screening Method to Detect Antimicrobial Volatiles from Metagenomic Clone Libraries

**DOI:** 10.3390/antibiotics9110726

**Published:** 2020-10-22

**Authors:** Franz Stocker, Melanie M. Obermeier, Katharina Resch, Gabriele Berg, Christina A. Müller Bogotá

**Affiliations:** 1Institute of Environmental Biotechnology, Graz University of Technology, Petersgasse 12/I, 8010 Graz, Austria; franz.stocker@tugraz.at (F.S.); m.obermeier@tugraz.at (M.M.O.); katharina_resch@gmx.at (K.R.); gabriele.berg@tugraz.at (G.B.); 2Austrian Centre of Industrial Biotechnology, Petersgasse 14/V, 8010 Graz, Austria

**Keywords:** volatiles, antimicrobials, functional metagenomics, high-throughput screening, *Sphagnum*, *Fusarium* spp.

## Abstract

The ever-growing spread of resistance in medicine and agriculture highlights the need to identify new antimicrobials. Microbial volatile organic compounds (VOCs) are one of the most promising groups of chemicals to meet this need. These rarely exploited molecules exhibit antimicrobial activity and their high vapour pressure makes them ideal for application in surface sterilisation, and in particular, in biofumigation. Therefore, we adapted the previously developed Two Clamp VOCs Assay (TCVA) to a new high-throughput screening for the detection of novel antifungal VOCs from metagenomic clone libraries. As a proof of concept, we tested the new high-throughput TCVA (htTCVA) by sourcing a moss metagenomic library against *Fusarium culmorum*. This led to the identification of five clones that inhibited the growth of mycelium and spores in vitro by up to 8% and 30% and subsequently, to the identification of VOCs that are potentially, and in part responsible for the clones’ antifungal activity. For these VOCs, the in vitro effect of the pure compounds was as high as 100%. These results demonstrate the robustness and feasibility of the htTCVA, which provides access to completely new and unexplored biosynthetic pathways and their secondary metabolites.

## 1. Introduction

Antimicrobial compounds have become an integral part of human medicine and farming. Consequently, the ever-increasing emergence and spread of antimicrobial resistance (AMR) threaten the advancement of modern medicine and pose a risk to food security [1,2]. To reduce the predicted 10 million AMR-related deaths each year by 2050 [3] and yield losses, which currently amount to about 30% per annum [2], new antimicrobial compounds are urgently needed, especially those with a novel mode of action [2].

Bacterial volatile organic compounds (VOCs) appear to be a promising molecule class, yet they have rarely been exploited to this end. These secondary metabolites, which are commonly fatty acid derivatives, terpenoids, aromatic, nitrogen or sulphur containing compounds possess high vapour pressure, which allows them to circulate in the atmosphere and in aqueous solutions [4]. Thus, VOCs serve as communication molecules that exert functions such as antimicrobial activity, above as well as below ground [5]. Between 50–80% of cultivated bacteria produce VOCs, whereby up to 80 different compounds are released per strain. Although overlaps occur, each strain produces a specific assemblage of VOCs, which in turn elicit species to species specific responses [5]. This exemplifies the wealth of bacterial VOCs and their potential in strategies for broad-spectrum as well as targeted pathogen management. Since the importance of the microbiome for human and plant health has been commonly recognised [6,7], where possible, targeted approaches are preferred over broad-spectrum agents to maintain microbiome integrity. Antimicrobial VOCs are generally useful for surface sterilisation, are they are particularly ideal for application in biofumigation to protect plants against fungal infestation. One third of bacteria have been found to produce fungistatic volatiles [8] and the antagonistic effect of bacterial VOCs on plant-pathogenic fungi has already been demonstrated [9,10,11,12]. Their mode of action involves reduced sporulation, mycelial growth inhibition and changes in morphology that range from disappearance of septae through to cell lysis [5].

Plant-associated microbiomes represent a promising bio-resource for antimicrobial VOCs, because communication and pathogen defence constitute the integral functions of plant-associated microbiomes that are essential for their host´s health [13]. To harness this potential and cost-effectively identify VOC-producing bacteria with antifungal activity from such bio-resources, we previously developed the Two Clamp VOCs Assay (TCVA) [10]. The TCVA has already been used to identify VOC-producing bacteria that antagonise plant-pathogenic fungi [14]. However, the majority of bacterial volatiles remains inaccessible as current methodologies only allow cultivation of ~5% of the bacterial diversity [15]. One way to circumvent cultivation-dependence lies in functional metagenomics [16]. By isolating metagenomic DNA directly from a complex microbial sample and cloning it into a surrogate host for heterologous expression, functional metagenomics can be utilised to access otherwise hidden pathways and metabolites. Efficient high-throughput screening (HTS) methods to make the screening of thousands of generated metagenomic clones for antimicrobial VOCs feasible are not in place. Thus, we established the first HTS method to this end by adapting the existing TCVA. Similar to the original assay, the high-throughput TCVA (htTCVA) set-up consists of two well plates separated by perforated silicon. In the self-contained chambers that are thereby created, the effect of single metagenomic clones or isolates on the opposite, plant-pathogenic fungus is monitored by means of fungal proliferation. Using 96-well plates and spores instead of 6-, 12- or 24-well plates and plaques of mycelium, the htTCVA facilitates a throughput of 13.000 test organisms per week, resulting in a 4 to 16-fold increase.

The aim of this study was to test the robustness and feasibility of the newly established htTCVA in respect to the identification of antifungal VOCs from large strain collections. Therefore, a moss metagenomic library was screened for antifungal VOCs. Previous work has provided a detailed understanding of the taxonomically and metabolically rich microbiome associated with *Sphagnum* mosses and its potential for the identification of antifungal compounds [17,18,19,20]. This potential was harnessed against *Fusarium culmorum*. This plant-pathogenic fungus produces mycotoxin, which is very economically relevant for the wheat sector because it lowers grain quality and yield and also raises concerns regarding food safety [21]. Our approach led to the successful identification of antifungal VOCs from uncultivable microorganisms, which inhibit the growth of *F. culmorum* spores and mycelium in vitro, thereby validating the robustness of the htTCVA.

## 2. Results

The focus of this study was to establish and validate a new HTS method for the identification of antifungal VOCs by exploiting metagenomic clone libraries. Such a library, generated using the microbiome of *S. magellanicum,* was screened for VOCs displaying antagonistic effects against the fungal plant pathogen *F. culmorum*. In this section the results of the three-step screening process are described. The most promising clones were then selected for further evaluation. The inhibition rates of the selected metagenomic clones were compared by analysing different versions of the Petri Dish VOCs Assay (PDVA). Additionally, VOC profiling was performed by headspace solid phase micro extraction gas chromatography-mass spectrometry (SPME GC-MS) to identify clone-specific VOCs (Figure 1).

### 2.1. Screening of Metagenomic Libraries for Antifungal VOCs

The initial screening of 18,124 metagenomic clones employing the htTCVA (Figure 1a and Appendix A) led to the identification of 376 metagenomic clones displaying volatile antagonistic activity against *F. culmorum*. The antagonistic effect of these positive tested clones could be confirmed for 27 clones during the second and third step of the strategy, the TCVA-based re-screening (Figure 1b and Appendix A). The five most active clones were then selected for downstream analysis (clone 130 F2, 131 B5, 131 E3, 131 F2, 172 E4).

The five clones were proven non-redundant by restriction enzyme digestion (data not shown) and de novo sequencing of the fosmid inserts (Appendix A). The hit rate of the functional metagenomic screening was calculated to be 118.1 Mb DNA hit^−1^.

### 2.2. PDVA: Determination of Growth Inhibition Rates

Determination of the relative growth inhibition rates (with respect to the empty vector control strain) (Figure 1c, Appendix A) of the five metagenomic clones showed a stronger inhibition efficacy against spores as compared to fungal mycelium (Figure 2a,b).

Five clones caused growth inhibition of the mycelium of *F. culmorum*, whereby 130 F2 and 131 E3 showed the best mean inhibition rates with 8% and 7%, respectively. Additionally, the metagenomic library clones 172 E4 and 131 B5 displayed a mean inhibition rate of 6% and 131 F2 showed a rate of 4%. *P. putida* KT2440, which was chosen as the reference strain due to the reported antagonistic activity of *P. putida* strains against *Fusarium* spp. [22], caused the highest inhibition rate with 16% (Figure 2a). The calculated inhibition rates are very close to statistical significance according to ANOVA with a *p* value of 0.05523.

When testing the antifungal effect against fungal spores, 131 E3 showed the highest inhibition rate with 28%, followed by 131 F2 with 26% and 131 B5 with 22%. All surpassed the positive reference *P. putida* KT2440, which had a rate of 19% (Figure 2b). 130 F2 and 172 E4 did not inhibit the growth of the fungal spores (data not shown). The inhibition rates were on the border of statistical significance according to ANOVA with a *p* value of 0.06814.

### 2.3. VOCs Profiling Through SPME GC-MS

SPME GC-MS-based VOC profiling of the five metagenomic library clones led to the identification of nine specific peaks, which were unique for the clone or highly up-regulated as compared to the control strain, *E. coli* EPI300 pCC2FOS (Appendix A). For eight of these, the respective compounds could be identified and confirmed using identical standards (Table 1). Thereby, three different approaches were employed for the analysis of VOCs, which resulted in the identification of the different compounds: single cultivation, co-cultivation with the fungus in one vial, and co-cultivation in separate vials.

When profiling the metagenomic library clones without prior co-cultivation with the fungus, the fatty alcohols 1-decanol and 1-undecanol were observed for 131 E3. Additionally, 2-phenoxyethyl acetate was found (Table 1). For the metagenomic clones 131 B5 and 131 F2 the same fatty alcohols, 1-decanol, 1-undecanol, and also 1-dodecanol were observed. In the case of the metagenomic library clone 130 F2, valeric acid and 1-decanol were detected (Table 1).

The second approach, the analysis of VOC-mediated interaction between the clones and the fungus by co-cultivation in a single vial, led to the detection of the sesquiterpene α-bisabolene and one unknown compound with a retention time of 23.587 min in library clone 131 E3 (Table 1).

To determine whether compounds that accumulated during co-cultivation got modified by the fungus, a second co-cultivation was performed using two vials. The monoterpene linalool and the methylated alkane 2,2,4,6,6-pentamethylheptane were detected by this approach in the co-cultivation of 130 F2 and 172 E4, respectively (Table 1).

### 2.4. In Vitro Verification of the Identified VOCs and Determination of Their Minimal Inhibitory Concentration

For the identified, potentially antifungal VOCs, the commercially available substances were tested in vitro to corroborate the SPME GC-MS findings and all showed an antifungal effect through their volatile interaction against *F. culmorum*. 

The highest antifungal effect was observed for linalool (RT 13.504 min) and valeric acid (RT 7.509 min) with 100%, followed by 1-decanol (RT 18.570 min) with 52%. For the other compounds the effects were determined as follows: 1-undecanol (RT 19.990 min) with 48%, 2-phenoxyethyl acetate (RT 20.236 min) with 25%, 1-dodecanol (RT 21.541 min) with 14%, α-bisabolene (RT 22.533 min) with 12% and 2,2,4,6,6-pentamethylheptane (RT 10.119 min) with 11% (Figure 3a). These inhibition values were statistically significant with a *p* value of 0.001702, according to Kruskal–Wallis.

To obtain a better understanding of the inhibition efficacy of the identified compounds, serial dilutions of all VOCs were tested against *F. culmorum* (Appendix A). The molar amount of the substances ranged between 0.1 and 933 µmol. The minimal inhibitory concentrations (MICs) for valeric acid and linalool as applied in the liquid form (150 µL), were determined to be 6.7 mM (1 µmol) and 1.7 M (250 µmol), respectively (Figure 3b). Due to technical limitations, it was not possible to quantify the real concentration of the substances in the gas-phase during the course of the cultivation. Assuming that the available headspace of the petri plate is 50 mL (80 mL total volume minus 30 mL agar) and total evaporation of the substance during cultivation, the maximal theoretical estimated concentration of these two substances in the gas-phase would account for 0.02 mM for valeric acid and 5 mM for linalool. For the remaining compounds 100% inhibition was not observed for the tested concentrations. Effective concentrations for partial inhibition (considered to be ≥10%) ranged between 1.6 and 6.2 M in the liquid form. Since it cannot be assumed with certainty that the whole molar amount of the substance evaporated in the gas-phase during cultivation, the results should be interpreted with caution.

### 2.5. Growth Inhibition of Different Fungi by 130 F2 and 131 E3

The inhibition potential of clones 130 F2 and 131 E3, which showed the highest efficacy against *F. culmorum* and produced two of the most effective antifungal VOCs, was tested against other phytopathogenic fungi. This was of interest to prove that the clones can inhibit other fungi and also that the assays allow the identification of clones that act antagonistically towards different fungi. No inhibitory effect was detected against *Rhizoctonia solani* and *Botrytis cinerea*. The inhibitory effect against *Fusarium verticilloides* with rates of 4% and 2% was not significant. In contrast, 130 F2 and 131 E3 showed a significant antagonistic effect according to Kruskal–Wallis (*p* value 130 F2 = 0.001328; 131 E3 = 0.01462) against *Sclerotinia sclerotium* with an inhibition rate of 21% and 48%, respectively (Figure 4).

## 3. Discussion

The successful identification of antifungal VOCs from a metagenomic library, which represents the first functional metagenomics screening for volatiles, demonstrates the robustness and feasibility of the newly established high-throughput screening method. It further confirms that functional metagenomics allow the identification of antimicrobial VOCs from uncultivable microorganisms. Therefore, the assay is not limited to large clone libraries and fungi as target organisms, but it is also applicable to culture collections and against bacteria and yeast. As shown by our approach, the htTCVA facilitates prospecting procedures for species-specific biological effects in a time and cost-effective manner. Given the described throughput of 13,000 clones per week presented here, it accelerates the screening by several fold as compared to existing assays.

The presented workflow consists of a three-step screening process that employs htTCVA as the initial step followed by the TCVA in the second and third step. As the assay assesses a strain’s entire volatilome and not individually produced VOCs, compounds of interest have to be singled out through analytics. Therefore, the three-step screening procedure is important to mitigate the analysis of false positives. The individual screening steps include pre-incubation of the fosmid clones. The resulting growth advantage should fit them against VOCs released by the fungus, but first and foremost, it enhances the hit rate as volatiles are predominantly produced and emitted during the stationary growth phase [29].

To validate the developed HTS method and the overall experimental design and workflow, a *Sphagnum* moss metagenomic library was screened against *F. culmorum*. Previous research has highlighted the great antifungal potential of the *Sphagnum* microbiome [18,30,31] to identify a number of bioactive clones. The results meet these expectations and corroborate our approach. From a total of 18,124 screened fosmid clones, five clones emitted antifungal VOCs. Eight VOCs were successfully identified through headspace SPME GC-MS and shown to be active against *F. culmorum* spores and mycelium in vitro, thus clearly demonstrating the effectiveness of the htTCVA in identifying antifungal VOCs.

The hit rate of 144.1 Mb DNA hit^−1^ resulting from the identification of five out of 18,124 metagenomic clones may seem low, yet meets the expectation for the employed methodology, which for instance can range between 2.7 Mb DNA hit^−1^ and 3979.5 Mb DNA hit^−1^ for enzyme screenings [32]. As functional metagenomics circumvent cultivation dependence by heterologous expression of metagenomic DNA in a surrogate host, the success rate is influenced by multiple factors that are inseparably linked to each other [33]. These include the host–vector system, size of the target gene, its abundance in the source metagenome and the efficiency of heterologous gene expression in a surrogate host. For the construction of a metagenome library, *E. coli* is usually used as a host [34]. *E. coli* is easy to handle microbiologically, grows fast and to high culture density and can be transformed superbly with recombinant vectors. Also, a broad range of efficient genetic tools, selection and expression systems are available [35,36]. However, there are problems with using *E. coli* for heterologous expression of metagenomic DNA. Since the metagenome is a complex mixture of genomes from a diverse array of microorganisms, the genetic machineries of *E. coli* often cannot recognise transcriptional and/or translational signals from metagenomic DNA and fail to decode certain genetic information therein [37].

The issue of heterologous expression in *E. coli* explains why the observed inhibition rates for the positive tested clones are rather low, yet close to statistical significance when using *F. culmorum* as test organisms. Additionally, the mechanisms that take place in the assay also play a role. The clones first need to fill the aerial space with the inhibitory VOCs to exactly the concentration that leads to an antifungal effect. Until that concentration is reached, the fungus proliferates normally at first, then shows signs of phenotypic stress response (e.g., change in color, less dense mycelium, etc.) and eventually stops growing. However, based on other published work that has used antifungal bacteria [38,39], the inhibitory effect of the metagenomic clones lie in the inhibitory range that can be expected for such assays.

For *S. sclerotium,* a statistically significant inhibition (*p* value < 0.05) was observed for the metagenomic clones 130 F2 and 131 E3. However, at this stage, it is not clear at what concentrations the clones emit the detected VOCs, like valeric acid or linalool, or what the MIC of these compounds is for *S. sclerotium*. In fact, it is very likely that better inhibition occurs through more than just one antifungal VOCs, as bacteria or in this case the clones, emit several, different VOCs [5], of which many could possess antifungal potential. Thus, the inhibitory efficacy can differ, so that one compound might cause the observed inhibition to a higher degree then other antifungal VOCs. Additionally, the inhibition rate of a given VOC depends on the specific resilience of the fungus to that particular compound [26,38,39].

The limitations of heterologous expression can be counteracted through alternative hosts, which could also be better for secretion of secondary metabolites. Nevertheless, it has been suggested that about 40% of the enzymatic activities may be easily recovered by random cloning in *E. coli* [37]. For maximised product identification, these factors should be carefully considered with regard to the desired product when designing a htTCVA approach.

Despite this limitation, functional metagenomics grants access to microorganisms that are not currently cultivable and their otherwise hidden genes and pathways, thus allowing the identification of truly novel secondary metabolites including antimicrobial VOCs. The established, highly efficient HTS thereby facilitates rapid bioprospecting of these largely unexplored bio-resources and the discovery of truly novel VOCs. This was reinforced by the identification of one metagenomic clone (131 E3), which contained a high number of unknown genes and produces an unknown VOC. The htTCVA in combination with screening of clone libraries, allows for faster gene identification compared to the evaluation of microbial isolates. This adds additional value to the identification of VOCs via functional metagenomics as not much is known about the mechanisms and genes involved in VOC biosynthesis. For instance, β-oxidation and fatty acid synthesis constitute common biosynthesis pathways for VOCs [40,41,42,43] and the respective genes were annotated on the fosmid inserts. In the case of the detected VOCs, valeric acid, 1-decanol, 1-undecanol, 1-dodecanol, and 2,2,4,6,6-pentamethylheptane might be produced during β-oxidation [42], or, except for the latter, during fatty acid synthesis [37].

Seven of the eight identified VOCs have been previously linked to antifungal activity. For instance, 2,2,4,6,6-pentamethylheptane was reported as component of antifungal ether extracts [24]. The natural sesquiterpene compound α-bisabolene, as well as the monoterpene linalool are components of essential oils [25,44]. Such oils have antifungal activity against *F. culmorum* [28]. In the case of 2-phenoxyethyl acetate, and to the best of our knowledge, there is no other scientific report that mentions its direct inhibition against fungi, making this the first report to describe its antifungal potential. Interestingly, the observed inhibition could occur due to compound transformation by the fungus. Hydrolases on the outer cell wall of *F. culmorum* can cleave C-O ester bonds, resulting in the production of phenolic compounds [45]. Phenolic compounds are able to inhibit fungal proliferation [46]. Compared to the clones, the determined MICs of the pure compounds seem high. Yet, such high concentrations have been reported before by Gao et al., who calculated the half maximal effective concentration (EC_50_) for benzhothiazole, anisole and 3-methylbutanal against *Colletotricum gloeosporioides* as 0.36 M, 0.59 M and 7.67 mM, respectively [38]; whereby *Fusarium* spp. are more resilient against VOCs [39]. Additionally, the determined MICs stem from the concentration of the applied liquid compounds, which would be expected to be higher than the concentration of the VOCs in the aerial space. As the quantification of the real and effective concentration of the substances for inhibition through the gas phase are challenging due to technical difficulties (e.g., compound enrichment during cultivation, no chance of sampling from petri plates), the theoretical concentrations were calculated. Considering this and already published work, the MICs presented here are well in line with the available literature.

As the identification of 2,2,4,6,6-pentamethylheptane and linalool during co-cultivation experiments indicates, compound modification through the fungus itself may lead to new or more potent antimicrobial VOCs. The active compound may also be produced as a response during VOC-mediated interaction, as exemplified in this study by α-bisabolene. This clearly shows that VOC identification depends on the cultivation method used prior to analytics and needs to be taken into account, since analysing only single-cultivated clones may miss promising compounds or entirely fail to identify antimicrobial VOCs.

Notably, limitations in product identification are not only related to the right analytics approach, but also to the set-up of the htTCVA, which may be associated with a high false positive and negative rate. This rate can be expected to be lower when testing isolates as compared to the functional metagenomics strategy pursued here. While the identification of truly novel natural products is one of the great advantages of this methodology, phenotypic instability of the metagenomic clones can occur due to the large insert vector. However, the assay is not designed to reach high reproducibility across the different testing steps, but to facilitate a high throughput for rapid identification of the most potent VOC producers.

## 4. Materials and Methods

### 4.1. Strains and Culture Conditions

Unless otherwise stated, the strains used in this study were cultured as follows: *Botrytis cinerea* PM14 (DSMZ 28931), *Fusarium culmorum* FC, *Rhizoctonia solani* AG-4, *Sclerotinia sclerotium* Goa11 and *Fusarium verticillioides* obtained from the strain collection of the Institute of Environmental Biotechnology (SCAM, Strain Collection of Antagonistic Microorganisms, Institute of Environmental Biotechnology, TU Graz, Austria) were cultivated on potato extract glucose bouillon (PDA; Carl Roth, Germany) agar plates at 20 °C. The fosmid clones in *E. coli* EPI300 pCC2FOS (Epicentre, Wisconsin, USA) were cultivated at 37 °C in Luria-Bertani (LB) medium supplemented with chloramphenicol (12.5 µg mL^−1^; Carl Roth, Germany) to maintain the vector. l-Arabinose (0.01% *w*/*v*; Sigma-Aldrich, Germany) was also added to the medium to induce a high copy number of the CopyControl fosmid pCC2FOS. *Pseudomonas putida* KT2440::TnRS48 pRS44, which contained a pCC2FOS derivative with pRS44 [47], was cultivated on LB medium amended with kanamycin (50 µg mL^−1^; Carl Roth, Germany) and tetracycline (10 µg mL^−1^; Carl Roth, Germany) at 30 °C.

### 4.2. Screening for Antimicrobial VOCs

A fosmid library with clones holding ~40 kb fragments of metagenomic DNA from the moss *Sphagnum magellanicum* sampled at the Alpine peat bog Pirker Waldhochmoor (N 46°37′38.66″, E 14°26′5.66″) was previously generated in *E. coli* EPI300 pCC2FOS (Epicentre, Wisconsin, USA) [30] and employed in this study. Using 96-well plates and the high-throughput Two Clamp VOCs Assay (htTCVA) as the initial step, 18,124 fosmid clones were screened against *Fusarium culmorum* spores. Positive clones were re-evaluated in two additional screening steps using the Two Clamp VOCs Assay (TCVA) with 12- and then 6-well plates. All screenings were carried out in duplicate (Figure 1a–c and Appendix A).

For the htTCVA as well as the TCVA, one well plate containing single library clones and one plate with fungal spores were assembled such that the top sides of each plate, separated by perforated silicon (0.2 mm width, 1 or 3 × 0.5 mm diameter holes for 96- well plates or 12- and 6-well plates), faced one another and were then fixed with two clamps. The spore suspension was prepared prior to usage by mobilising spores from 4-day old fungal mycelium into 10 mL sterile dH_2_O by scraping the mycelium with a sterile drigalski spatula. Spores were subsequently separated from hyphae by passing the suspension through sterile mull and the spore concentration was adjusted to 100 spores µL^−1^. The suspension was stored at 4 °C for up to 48 h. For cultivation of the library clones, all wells of the 96-, 12- and 6-well plates were filled with 70 µL, 1 mL or 3 mL of LB agar (12.5 µg mL^−1^ chloramphenicol, 0.01% *w*/*v*
l-arabinose) and inoculated with single metagenomic clones using 10 µL, 30 µL or 70 µL directly taken from the 25% (*v*/*v*) glycerol stocks, respectively. The empty vector strain *E. coli* EPI300 pCC2FOS served as negative control. As *P. putida* strains are known to inhibit *Fusarium* spp. [22], *P. putida* KT2440::TnRS48 pRS44 was used as a positive reference strain. The positive reference strain contained the pRS44/TnRS48 cloning system, a derivative of pCC2FOS [47], to rule out any impact of the pCC2FOS fosmid cloning system. After pre-incubation of the library containing plates at 37 °C for 24 h, 30 µL, 150 µL or 500 µL bacillol (Bacillol AF, BODE Chemie GmbH; composite: 1-propanol (CAS: 71-23-8), 2-propanol (CAS: 67-63-0), ethanol (CAS: 64-17-5)) were added into dedicated wells as additional positive control. The counterpart plates filled with the same volumes of PDA agar were then inoculated with 10 µL, 20 µL or 30 µL of spore suspension (for 96-, 12- and 6-well plates, respectively) and the plates assembled as described above. The set-up was incubated at 20 °C for 5 days. The turbidity of the agar, judged by visual inspection, indicated whether *F. culmorum* was inhibited or not.

The hit rate in base pairs DNA per hit was calculated based on the overall amount of 708.6 Mb DNA tested as deduced from the number of tested clones and the average fosmid insert size of 39.1 kb.

### 4.3. Quantification and Validation of the Antagonistic Effect

To determine the inhibition efficacy of identified clones against spores and fungal mycelium, overnight cultures of single metagenomic clones and the control strains were diluted to an OD_600_ of 0.8 and 100 µL plated onto LB agar (12.5 µg mL^−1^ chloramphenicol, 0.01% *w*/*v*
l-arabinose) plates. Petri dishes (92 × 16 mm, 80 mL max. volume) were used to this end. After pre-incubation of the clone containing petri plates for 24 h, the counterpart plates were prepared. Therefore, a plaque of mycelium (Ø 8 mm) or 5 µL of spore suspension was transferred on the centre of an PDA agar plate and the plates were assembled face-to-face and sealed with parafilm. Tests were carried out at a sample size of *n* = 5 for mycelium and *n* = 3 for spores, using the empty vector strain *E. coli* EPI300 pCC2FOS as the negative control. *P. putida* KT2440::TnRS48 pRS44 served as a positive reference to test if growth inhibition was observed with this assay. The assembled assays were positioned such that plaque-containing plates were upside-down, while plates holding spores were on the bottom (see Appendix A). These were incubated at 20 °C for 5 days or until the first mycelium reached the rim of the petri dish. To determine the inhibition rates, the diameter (d) of the circular grown mycelium was measured for all plates in mm (Appendix A) and the relative inhibition rate (%) of the clone (or *P. putida*) was determined against the empty vector control strain. Mean values and standard deviations for each compound were calculated for all determined relative inhibition rates. An ANOVA test was conducted with the parametric samples and a Kruskal–Wallis test was used with the non-parametric samples to statistically validate the observed inhibitory effect.

To corroborate the inhibitory effect of the detected VOCs against fungal mycelium, 150 µL of the pure, purchased compounds, 1-undecanol (4.77 mol L^−1^; CAS number 112-42-5; 99.0% GC), 1-decanol (5.13 mol L^−1^; CAS number 112-30-1; 98.0% GC), 1-dodecanol (4.38 mol L^−1^; CAS number 112-53-8; 98.0% GC), 2-phenoxyethyl acetate (6.22 mol L^−1^; CAS number 103-45-7; 99.0% GC) (Sigma-Aldrich, Germany), α-bisabolol (3.35 mol L^−1^; CAS number 515-69-5; >80.0% GC), linalool (5.41 mol L^−1^; CAS number 78-70-6; >96.0% GC) and 2,2,4,6,6-pentamethylheptane (4.31 mol L^−1^; CAS number 13475-82-6; >98.0% GC) (Tokyo Chemical Industry Co., LTD, Japan GC), and valeric acid (9.14 mol L^−1^; CAS number 109-52-4; 100% GC) (Merck, Germany), were pipetted on a polytetrafluoroethylene (PTFE)-lined silicon rubber septum (LaPhaPack, Germany) placed in the centre of the petri dish lid. The sole PTFE-lined silicon rubber septum served as negative control. The counterpart plate holding a plaque of mycelium was prepared as described above and was placed upside down on the lid. The assembled petri dish was sealed with parafilm (Appendix A). Three pieces of paper towel (5 mm × 5 mm) soaked with 500 µL bacillol served as positive control in addition to the reference strain *P. putida* KT2440::TnRS48 pRS44. The available headspace of the petri plate, in which the tested substance could evaporate was estimated to have a volume of approximately 50 ± 5 mL. Tests were carried out in triplicate. The assembled assays were positioned such that plaque-containing plates were upside-down and with the rubber septa holding the lid on the bottom. Incubation, measurements, calculations of the relative inhibition rates and their statistical validation were performed as described above.

In the same way, serial dilutions of the compounds in DMSO for a final volume of 150 µL were tested against fungal mycelium to determine the MICs. Sole DMSO served as negative control. The molar amounts of the tested substances were in the range of 0.1 to 933 µmol (refer to Appendix A for details).

### 4.4. Qualitative Detection of VOCs by SPME GC-MS

Headspace solid phase microextraction gas chromatography-mass spectrometry (SPME GC-MS) was used to measure the volatilome of active library clones. Prior to the measurement, library clones and the empty vector control strain were streaked directly from glycerol stocks onto a 7 mL LB slope agar inside a 20 mL headspace vial (75.5 × 22.5 mm, Chromtech, Germany). The vials were closed with a cotton plug and covered with tin foil, and were incubated for 24 h and then sealed with magnetic crimp caps (BGB Analytik Vertrieb GmbH, Germany) with a PTFE-lined silicon rubber septum (LaPhaPack, Germany) followed by another two hours of incubation at 20 °C (Figure 1c and Appendix A).

To analyse whether other relevant VOCs were produced during co-cultivation of clones and fungus, two different approaches were undertaken. For co-cultivation in the same vial, 7 mL of LB agar was spread across the vial´s wall. The clones were streaked onto the agar and the vials were closed with cotton plugs and covered with tin foil, and incubated for 24 h. A plaque from a 5-day old fungal mycelium was then placed on the bottom of the vial, which was left free from LB agar and the vial was closed and covered (Appendix A). After 24 h incubation at 20 °C the vials were sealed as described above and incubated for another two hours at 20 °C before measurement. 

For co-cultivation in separated vials, the clones were cultivated on slope agar as described above. After the 24 h pre-incubation, the vial was connected head-to-head to a second head space vial filled with 20 mL PDA slope agar and containing plaque from a 5-day old fungal mycelium. The connected vials were sealed with parafilm and incubated for 24 h at 20 °C (Figure 1c and Appendix A). Afterwards, the vials containing the fungus were sealed with magnetic crimp caps with PTFE-lined silicon rubber septum and incubated for an additional 2 h at 20 °C for enrichment of VOCs, before measurement.

The profiling was performed with an automated sampler and a 50/30 μm Divinylbenzen/CarboxenTM/Polydimethylsiloxane 2 cm Stableflex/SS fiber (Supelco, PA, USA), and a GC 7890A combined with a quadrupole MS 5974C (Agilent Technologies, Germany). After enrichment of the volatile compounds for 30 min at 30 °C, samples were run through a (5% phenyl)-methylpolysiloxane column, 60 m × 0.25 mm i.d., 0.25 μm film thickness (DB-5MS; Agilent Technologies, Germany) followed by electron ionisation (EI; 70 eV) and compound detection (mass range 25 to 350 u). The helium flow rate was set to 1.2 mL min^−1^ and the inlet temperature set to 270 °C. The applied temperature gradient was as follows: 40 °C for 2 min, 40–110 °C with 5 °C min^−1^, 110–280 °C with 10 °C min^−1^, 280 °C for 3 min.

The VOC profiles of the metagenomic clones were compared to that of the control strain to identify unique peaks. The compounds were identified by matching their spectra and retention times against the NIST Mass Spectral Database 08 entries. Identification was verified using commercially available purchased reference substances, dissolved in dimethyl sulfoxide (CAS number 67-68-5; 100% GC (Sigma-Aldrich, Germany) to a concentration of 0.1 mol L^−1^. Ten μl of the diluted reference substance were transferred into sterile 20 mL headspace vials, which were sealed with magnetic crimp caps with a PTFE-lined silicon rubber septum and incubated for two hours for evaporation and enrichment in the headspace prior to SPME GC-MS analysis. Analyses were conducted in triplicate.

### 4.5. Data Accessibility

The nucleotide sequences of the fosmid inserts can be found at Genbank (BioProject ID: PRJNA662101, BioSample ID SAMN16076770) under the accession no. MW000465-MW000469.

## 5. Conclusions

Due to the great adaptability of the presented HTS, a variety of culture collections and metagenomes can be screened against fungi, including yeast, and bacteria. Therefore, the htTCVA represents a feasible and promising means for the rapid execution of large screening campaigns. To fully exploit this potential, the new HTS should ideally be coupled to the ethnobotanical approach [48]. We anticipate that natural product research based on the ethnobotanical approach and using the established htTCVA could lead to the identification of truly novel lead compounds for the medical and agricultural sector.

## Figures and Tables

**Figure 1 antibiotics-09-00726-f001:**
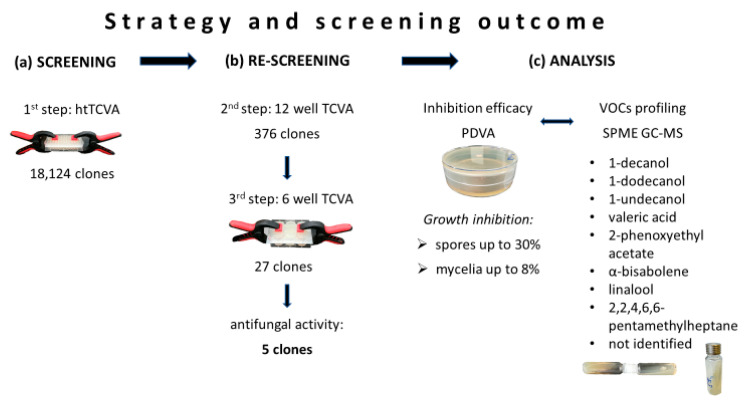
Schematic representation and results of the three-step screening process for antifungal volatile organic compounds (VOCs). (**a**) Initial step to identify metagenomic clones that emit antifungal VOCs employing the high-throughput Two Clamp VOCs Assay (htTCVA). (**b**) Two Clamp VOCs Assay (TCVA) as second and third step to confirm positive tested clones. (**c**) Evaluation of identified clones including the determination of the inhibition efficacy through the Petri Dish VOCs Assay (PDVA), and profiling VOCs to identify compounds of interest through headspace solid phase micro extraction gas chromatography-mass spectrometry (SPME GC-MS).

**Figure 2 antibiotics-09-00726-f002:**
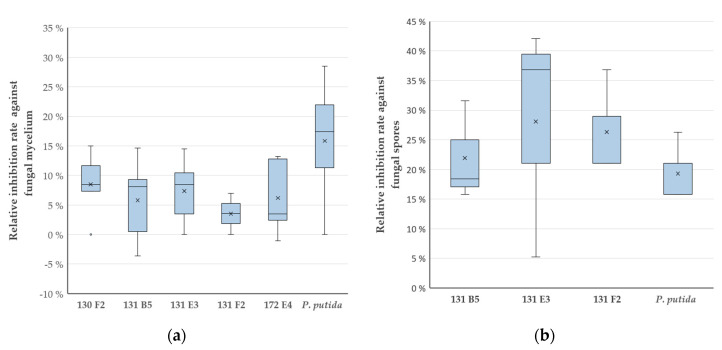
Effect of antagonistic fosmid clones from a moss metagenomic library on *F. culmorum*. Relative inhibition rates against fungal mycelium (**a**) and spores (**b**) of metagenomic clones and the reference strain *Pseudomonas putida* KTT2440 as compared to the empty vector control strain *E. coli* EPI300 pCC2FOS are displayed in percent. The box plots show the median (-), the mean (×), the outliers (°) and the standard deviation (whiskers) of samples; sample size *n* = 3.

**Figure 3 antibiotics-09-00726-f003:**
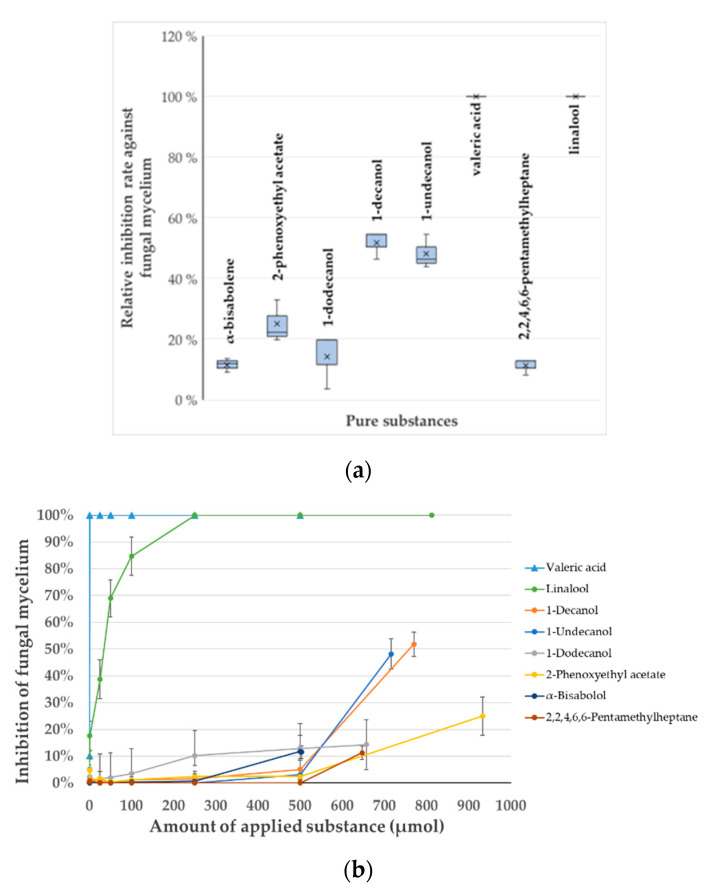
Antagonistic effect of pure compounds on *F. culmorum* as determined by the Petri Dish VOCs Assay (PDVA). (**a**) Relative inhibition rates of detected VOCs as compared to an empty polytetrafluoroethylene (PTFE)-lined silicon rubber septum against fungal mycelium are displayed in percent. The box plots show the median (-), the mean (×) and the standard deviation (whiskers) of samples; sample size for fungal mycelium *n* = 5 and for fungal spores *n* = 3. (**b**) Growth inhibition of fungal mycelium in percent for the identified VOCs at different concentrations as mean values with standard deviation, *n* = 3.

**Figure 4 antibiotics-09-00726-f004:**
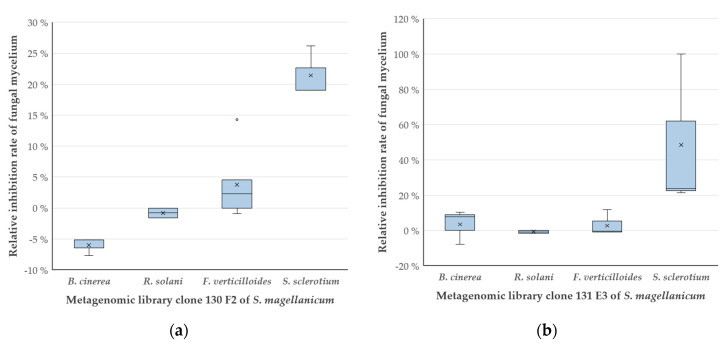
Effect of antagonistic fosmid clones from a moss metagenomic library on phytopathogenic fungi. Relative inhibition rates (in percent) of clone 130 F2 (**a**) and 131 E3 (**b**) as compared to the empty vector control strain *E. coli* EPI300 pCC2FOS against fungal mycelium from *Rhizoctonia solani, Botrytis cinerea, Fusarium verticilloides* and *Sclerotinia sclerotium*. The box plots show the median (-), the mean (×), the outliers (°) and the standard deviation (whiskers) based on PDVA tests with samples; sample size *n* = 3.

**Table 1 antibiotics-09-00726-t001:** VOCs produced by the metagenomic clones. List displaying the detected VOCs, retention times, the producing library clone and previously reported antifungal activity. VOCs, which were detected as interaction compounds upon co-cultivation of the clones with the fungus in a single vial (1) or in separate vials (2) are marked.

Volatile Organic Compounds	Retention Time (min)	Library Clone	Reported Antifungal Activity
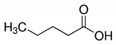 Valeric acid	7.509	130 F2	*Fusarium oxysporum*[23]
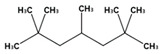 2,2,4,6,6-Pentamethylheptane(1)	10.119	130 F2; 172 E4	*Candida albicans, Cryptococcus neoformans, Trichophyton rubrum, Aspergillus fumigatus*[24]
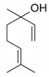 Linalool(2)	13.504	130 F2; 172 E4	*Fusarium spp.*[25]
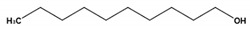 1-Decanol	18.570	130 F2; 131 B5; 131 E3; 131 F2	*Fusarium oxysporum*[26]
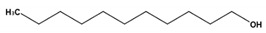 1-Undecanol	19.990	131 B5; 131 E3; 131 F2	*Penicillium chrysogenum, Aspergillus niger, Mucor mucedo, Trichophyton mentagrophytes*[27]
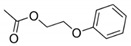 2-Phenoxyethyl acetate	20.236	131 E3	none
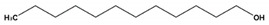 1-Dodecanol	21.541	131 B5; 131 F2	*Fusarium oxysporum*[26]
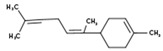 α-Bisabolene(2)	22.533	131 E3	*Fusarium spp.*[28]
Not identified(2)	23.587	131 E3	----

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
