# Peer review of "A New High-Throughput Screening Method to Detect Antimicrobial Volatiles from Metagenomic Clone Libraries"

_antibiotics, 2020, doi:10.3390/antibiotics9110726_

Round 1
Reviewer 1 Report
The study of Stocker et al., aims to provide a new high-throughput screening method to detect antimicrobial volatile compounds. The proposed method has been validated by sourcing a moss metagenomic library against Fusarium culmorum.
The topic is interesting, since allow the simultaneous screening of thousands of isolates and the fast identification of the best VOC producers for potential biotechnological applications.
The work is well-written, the experimental design roust and results clearly presented and properly discussed.
A section on the statistical analysis must be added.
In the in vitro verification of the identified VOCs pure compounds have been tested. Maybe the antifungal effect it should have been tested at the same concentrations produced by the clones to better compare the results.
Why, for in vitro verification of the identified VOCs and growth inhibition of different fungi it has been tested the relative inhibition rates against fungal mycelium and not also fungal spores?
L22 and 83. I am not agree on the use of in vivo. In my opinion, all the assays reported in this study are performed using in vitro conditions.
L89. Define the genera, the first time you mention it in the text
L267-268. Specify the strain for each fungal species used in the study
L351. Change mL to ml.
Author Response
REVIEWER 1
The study of Stocker et al., aims to provide a new high-throughput screening method to detect antimicrobial volatile compounds. The proposed method has been validated by sourcing a moss metagenomic library against Fusarium culmorum.
The topic is interesting, since allow the simultaneous screening of thousands of isolates and the fast identification of the best VOC producers for potential biotechnological applications.
The work is well-written, the experimental design roust and results clearly presented and properly discussed.
A section on the statistical analysis must be added.
- Authors´ response: We value this suggestion and have conducted and implemented such an analysis; see line 119-120, 124-125 and 178-179 in the results and line 349-352 in the methods section.
In the in vitro verification of the identified VOCs pure compounds have been tested. Maybe the antifungal effect it should have been tested at the same concentrations produced by the clones to better compare the results.
- Authors´ response: Testing the antifungal effect of the pure compounds at the same concentrations as the original producer strain would for sure be interesting. As pointed out in the above raised suggestion, we worked with clones. As per that, the genes of interest are not in the metabolic background of the original producer strain, but in a synthetic background and the natural physiological concentration cannot be determined. More importantly, the aim of our approach was to establish a methodology that allows the identification of antifungal VOCs for biotechnological application. In that sense, the verification (Do the identified compounds inhibit the fungus?) served as the final step to proof and validate the concept of our established method (Can we identify antifungal compounds that way). Rather than focusing at physiological concentrations of the producer strains, especially in light of the biotechnological applicability (spray application of the compound, not the use of the clones as a kind of biocontrol agent), we believe it is of greater interest to understand more about the inhibitory concentrations of the compounds. Therefore, we conducted further experiments (PDVA assays) with serial dilutions to determine the range that approximates the minimal inhibitory concentrations of the identified compounds, see Table S1 and line 166-170 (main text). We hope this complies with the above raised suggestion.
Why, for in vitro verification of the identified VOCs and growth inhibition of different fungi it has been tested the relative inhibition rates against fungal mycelium and not also fungal spores?
- Authors´ response: It was of primary interest to generally verify that the identified compounds inhibit the fungus in order to demonstrate the robustness and feasibility of the established method, rather than exploring in details the inhibitory effect itself. To do so, we chose to only test against the mycelium for which we observed lower inhibition efficacy than for the spores. That made it a more interesting target to verify the effect. Additionally, fungicides are often applied when the infestation on the field is discovered, meaning after mycelium has been formed on plants. Therefore, it was of greater interest to test the inhibitory effect of the pure compounds against mycelium rather than the spores.
L22 and 83. I am not agree on the use of in vivo. In my opinion, all the assays reported in this study are performed using in vitro conditions.
- Authors´ response: We agree that we may have used the term “in vivo” out of its true context in an effort to facilitate easy differentiation between the different tests/experiments. We have, thus, changed our wording and now use “in vitro” as suggested; refer to line 23 and 83.
L89. Define the genera, the first time you mention it in the text
- Authors´ response: Fusarium culmorum was first mentioned and introduced at the end of the introduction in line 79. We therefore kept the abbreviation in line 89.
L267-268. Specify the strain for each fungal species used in the study
- Authors´ response: we included the strain specifications in the methods section, line 287-289.
L351. Change mL to ml.
- Authors´ response: Thanks for pointing that out. It now reads “ml” in line 381.
Reviewer 2 Report
The article by Bogota entitled "A new high-throughput screening method to detect antimicrobial volatiles from metagenomicclone libraries", in my opinion, does warrant acceptance for publication in antibiotics.
The authors constructed metagenomic clone libraries in order to detect volatile organic compounds. Also, they evaluated their antifungal activity. The manuscript is well-written and an interesting topic for readers in antibiotics.
Comments
- line 13, the authors mentioned that “A promising group of chemicals to meet this need are microbial volatile organic compounds (VOCs).” I believe that it is overstatement. Only VOCs gives the impression that it is promising. Secondary metabolites (NON-VOCs) will also provide attractive natural products. Please consider it.
- Why did the authors test antifungal activity against only Fusarium culmorum? Other fungal must be interesting.
- In terms of activity, I do not think that VOCs will be promising compounds in the future. Because they are not functionalized structures and their structures are very simple. Is there a candidate compound that will become a promising drug?
- Line 35, 10 Mio. should be 10 million.
- Figure 1, it is low resolution. Please remake it.
- Line 384, XY-XY. What is it?
- Line 408, ref 3, Please add pages.
End.
Author Response
REVIEWER 2
The article by Bogota entitled "A new high-throughput screening method to detect antimicrobial volatiles from metagenomicclone libraries", in my opinion, does warrant acceptance for publication in antibiotics.
The authors constructed metagenomic clone libraries in order to detect volatile organic compounds. Also, they evaluated their antifungal activity. The manuscript is well-written and an interesting topic for readers in antibiotics.
Comments
- line 13, the authors mentioned that “A promising group of chemicals to meet this need are microbial volatile organic compounds (VOCs).” I believe that it is overstatement. Only VOCs gives the impression that it is promising. Secondary metabolites (NON-VOCs) will also provide attractive natural products. Please consider it.
- Authors´ response: We fully agree that non-VOCs secondary metabolites represent highly promising natural products too. It is, hence, not our intention to portray that only VOCs are promising compounds. In our opinion “A promising group” does not mean it is the only promising group, but we understand that this can be perceived differently depending on the reader and so we rephrased the sentence, see line 13.
- Why did the authors test antifungal activity against only Fusarium culmorum? Other fungal must be interesting.
- Authors´ response: Indeed, other fungi are of interest. However, our focus was to establish an easy, cheap and fast method to identify new antifungal VOCs from large clone or isolate collections. As this was the priority and aim of our work, we chose culmorum as “model species” to demonstrate the feasibility and robustness of our method, that – as we envision – will be used in the future to identify antifungal VOCs against all the other interesting fungi.
As there obviously exists interest in other fungi, we embarked a bit on testing other fungal species by testing two clones against Rhizoctonia solani, Botrytis cinerea, Fusarium verticilloides and Sclerotinia sclerotium, see Fig. 4, line 167ff. This was to showcase that the method a) works for other fungi and b) one identified clone/isolate can also inhibit more than one fungus which makes it of greater biotechnological interest.
- In terms of activity, I do not think that VOCs will be promising compounds in the future. Because they are not functionalized structures and their structures are very simple. Is there a candidate compound that will become a promising drug?
- Authors´ response: Thanks for raising this concern. VOCs are so far rarely explored and exploited compounds. This already brings an interesting aspect, as they represent new chemical structure that have not yet been used to ward of pathogens for instance on crop fields. So, the pathogens were not yet exposed to them; at least not at concentrations higher than what naturally is emitted by various organisms. The focus of our study lies mainly on agricultural applicability of antimicrobial volatiles. In this regard, and not directly linked to medical applications, we believe that antimicrobial volatiles are easier to apply (e.g. bio-fumigation), cost-effective and more readily accessible (than more functionalized, more expensive, structures). More importantly, our own research shows that these overlooked compounds do have potential: Kusstatscher, P., Cernava, T., Liebminger, S. et al. Replacing conventional decontamination of hatching eggs with a natural defense strategy based on antimicrobial, volatile pyrazines. Sci Rep 7, 13253 (2017). https://doi.org/10.1038/s41598-017-13579-7
Many terpenes show antibacterial and/or antifungal activity, as reviewed in:
Schulz-Bohm, Kristin et al. “Microbial Volatiles: Small Molecules with an Important Role in Intra- and Inter-Kingdom Interactions.” Frontiers in microbiology vol. 8 2484. 12 Dec. 2017, https://doi.org/10.3389/fmicb.2017.02484
The discovery of novel antimicrobial volatiles and its applicability is a new field of research, as evidenced by the antibacterial compound schleiferon A/B:
Kai, M., Effmert, U., Lemfack, M.C. et al. Interspecific formation of the antimicrobial volatile schleiferon. Sci Rep 8, 16852 (2018). https://doi.org/10.1038/s41598-018-35341-3
Many essential oils (also the vapor phase = VOCs mixtures) are well-known for their antimicrobial activity and could be applied for indoor desinfection:
Puškárová, A., Bučková, M., Kraková, L. et al. The antibacterial and antifungal activity of six essential oils and their cyto/genotoxicity to human HEL 12469 cells. Sci Rep 7, 8211 (2017). https://doi.org/10.1038/s41598-017-08673-9
- Line 35, 10 Mio. should be 10 million.
- Authors´ response: The change was made as suggested, see line 35.
- Figure 1, it is low resolution. Please remake it.
- Authors´ response: Thanks for bringing this to our attention. Figure 1 was improved.
- Line 384, XY-XY. What is it?
- Authors´ response: “XY-XY” were merely place holders for the accession numbers of the sequences until the sequences are approved and available at GenBank. The accession no. are available now and this information was included, see line 413-414.
- Line 408, ref 3, Please add pages.
- Authors´ response: Pages were added, see line 439.
Reviewer 3 Report
The paper by Stocker et al. describs screening of volatiles produced by a metagenomic library of clones from moss-associated microbiota for their ability to inhibit fungal growth. The approach is interesting, but the way the data are presented leaves doubts whether the results are meaningful.
First, the effect is relatively small (between 3 and 15% inhibition, Fig. 2) and no statistical analysis is provided to demonstrate if these differences are significant. Considering that only <10% of initial hits were validated in the second pass (27 out of 376), it raises concern about overall reproducibility of the data.
Second, not enough experimental details are given for the inhibition rate measurements. “To determine inhibition rates the differences of the radius of the mycelium were measured in comparison to the empty vector control strain”. But the data (e.g. in Fig. 2) are presented as relative inhibition rates. I would like to see the raw measurements (as well as pictures of the plates, from which these measurements were taken) to determine whether 5% difference is experimentally distinguishable. What was the uncertainty of the measurement? This goes back to statistical significance of these small inhibition rates.
Third, for testing synthetic standards, 150 ul of pure compounds were used. How does it compare with the concentration of these compounds in the gas phase of bacterial clones? I suspect that 150 ul of pure compounds will result in much gas concentrations. If so, are these experiments really comparable? Yes, these compounds may inhibit fungal growth at high concentrations, but are they physiologically relevant?
Author Response
REVIEWER 3
The paper by Stocker et al. describs screening of volatiles produced by a metagenomic library of clones from moss-associated microbiota for their ability to inhibit fungal growth. The approach is interesting, but the way the data are presented leaves doubts whether the results are meaningful.
First, the effect is relatively small (between 3 and 15% inhibition, Fig. 2) and no statistical analysis is provided to demonstrate if these differences are significant. Considering that only <10% of initial hits were validated in the second pass (27 out of 376), it raises concern about overall reproducibility of the data.
- Authors´ response: Regarding the small inhibition rate of 3 – 15%. We agree that this appears small at first. However, when taking the bigger picture into consideration the inhibition rates are reasonable. We demonstrated the robustness and feasibility of our newly established method through functional metagenomics or in other words by using metagenomic clones (we did so, to prove that VOCs can be identified from uncultivable bacteria through functional methodology, which has not been tested/published before). This was of interest as functional metagenomics allows the identification of truly novel compounds through heterologous expression of environmental DNA in a surrogate host, such as coli. The used vector system is a low copy fosmid which is about 50 kb in size once the environmental DNA insert has been cloned into the vector. It has to be assumed that any product or biosynthetic route encoded by the environmental DNA insert is expressed in low amount, so that huge inhibitions rates are not to be expected.
Regarding the statistical analysis. We value this suggestion and have implemented such an analysis, see line 119-120, 124-125 and 178-179 in the results, 229-232 in the discussion and line 349-352 in the methods section.
Regarding the questioned reproducibility. Our method is not designed to reach a high reproducibility across the different testing steps, but to rapidly identify the most promising clones/isolates for potential biotechnological application. It is true, that only a low number of clones made it through all the testing stages of the method. In order to rapidly identify the most promising clones the first testing step occurs in 96 well plates knowingly that the very small set up/well chamber could lead to false positives and also false negatives because the conditions are not ideal for the fungus to grow or for the clone to fully exert its effect. Yet, it is especially this step that facilitates the fast pace of testing that represents a key element for natural product discovery and thus is needed when thousands of clones or isolates are to be tested. Because of that, the method includes two additional testing steps in 6 well plates and subsequently the determination of the inhibition efficacy in petri dishes to ensure that only the most promising clones/isolates (those with the best reproducibility and strongest inhibitory effect, on which ultimately the greatest interest lies) are analysed by SPME GC-MS measurements. We briefly attended to this aspect in the discussion of the original manuscript and have now added an additional sentence to further clarify the limitations and advantages of the established method; refer to line 269 – 276 of the revised manuscript. Prior to that, the issue with false positives has been mentioned to explain why the assay features three testing steps (1 x htTCVA and 2 x TCVA); see line 199 – 203.
Second, not enough experimental details are given for the inhibition rate measurements. “To determine inhibition rates the differences of the radius of the mycelium were measured in comparison to the empty vector control strain”. But the data (e.g. in Fig. 2) are presented as relative inhibition rates. I would like to see the raw measurements (as well as pictures of the plates, from which these measurements were taken) to determine whether 5% difference is experimentally distinguishable. What was the uncertainty of the measurement? This goes back to statistical significance of these small inhibition rates.
Authors´ response: We appreciate pointing out that there exists greater interest in a more detailed description of the measurements. We attended to this concern by including the requested data in the method section (lines 341-352) and in form of a new figure showing exemplarily the comparison of the control plate and a plate containing one metagenomic clone, see Supplementary Figure S3. Additionally, the requested statistical analysis has been conducted to answer to concerns about uncertainty as stated above (line 119-120, 124-125 and 178-179 in the results and line 349-352 in the methods section).
Third, for testing synthetic standards, 150 ul of pure compounds were used. How does it compare with the concentration of these compounds in the gas phase of bacterial clones? I suspect that 150 ul of pure compounds will result in much gas concentrations. If so, are these experiments really comparable? Yes, these compounds may inhibit fungal growth at high concentrations, but are they physiologically relevant?
- Authors´ response: Thanks for raising this point. The concentration of the compound in the gas phase in the case of 150 µl pure compound may be indeed higher than the gas concentration reached by the clones. Yet, we do not share the opinion that these experiments (clone vs pure compound) have to be comparable nor that the concentrations of the pure compounds have to be tested at physiologically relevant concentrations. It would be interesting to test physiological concentrations for isolates and in an environmental/ecological context to determine the degree of inhibition reached by isolates in the wild. Our method does not concern ecological aspects. It focuses on biotechnological application of the identified compounds. Furthermore, we test metagenomic clones. As of that we don´t think that testing physiological concentrations is relevant because we test an coli strain that carries a 40 kb DNA insert of environmental origin and is, hence, not the original host strain from which the piece of environmental DNA was extracted.
However, we agree that further experiments that explore the compounds´ effects at lower concentrations are a valuable addition to the paper. So rather than focusing at physiological concentrations of the surrogate producer strains, especially in light of the focus of this paper – biotechnological applicability – we conducted further experiments (PDVA assays) with serial dilutions to determine the range that approximates the minimal inhibitory concentrations of the identified compounds, see the results section lines 166-170, methods section lines 365-370, and Table S1 (in Supplementary). We hope this complies with the above raised suggestion.
Round 2
Reviewer 3 Report
Based on the authors’ response to my previous comments, I must recommend this paper to be rejected. First, presented data are not statistically significant (p > 0.05). While I’m not a fan of p-values, when this is combined with other points below, it’s quite disturbing. Second, the inhibition effects are small. I appreciate that the authors included an image of the plates to demonstrate how they measured growth inhibition (Fig. S3). I’m sorry, but this image is not convincing, at least to me. If this is the best the authors can show, I’ve serious doubts about the results. Finally, the pure compounds were tested at non-physiological concentrations. By physiological, I don’t mean ecologically relevant, but rather concentrations that are close to the experiment when a metagenomic bacterial clone inhibited fungal growth. If one is to use the experiment with pure compounds to prove that it is these molecules that cause inhibitory effects, the concentrations should be reasonable. I would argue that at high enough concentrations, almost everything becomes toxic. With all this combined, I’m not sure that the results presented in this manuscript are publishable and not artefacts.
Author Response
Comments of Reviewer 3:
Based on the authors’ response to my previous comments, I must recommend this paper to be rejected. First, presented data are not statistically significant (p > 0.05). While I’m not a fan of p-values, when this is combined with other points below, it’s quite disturbing.
- Authors ´ response: We share the reviewer´s opinion about p-values. While they are relevant, the strict fixation on >0.5 may not be the most adequate for certain areas. Especially when working with living organisms like fungi and bacteria. Even when strictly adhering to ever the same culturing conditions (as done also in this study), fluctuation occurs. While we almost every time observe an inhibition, it´s efficacy varies. Just simply having slightly more dead cells in the prepared bacterial cell suspension or having more biomass in the 5mm plaque of fungal mycelium could already make a difference. We are, therefore, convinced that our results are true results and no artefacts.
Second, the inhibition effects are small. I appreciate that the authors included an image of the plates to demonstrate how they measured growth inhibition (Fig. S3). I’m sorry, but this image is not convincing, at least to me. If this is the best the authors can show, I’ve serious doubts about the results.
- Authors ´ response: Based on our extensive experience with such assays and from published literature, we can say that this is in the normal inhibitory range that can be expected by such assays. We refer to the publications of Yuan et al. wherein the Fusarium oxysporum and to Gao et al. wherein antifungal inhibition against different fungi were tested. Both publications include pictures where the different antifungal effects can be observed. It has to be kept in mind that both groups worked with antifungal bacteria instead of metagenomic clones (Yuan et al., 2012; Gao et al., 2018). We included a further image of the assay plate for one clone exemplarily in Supplementary Fig. S3.
It is also important to consider the mechanisms that take place in the assay. When both plates are assembled and sealed the clones first need to slowly fill the air with the inhibitory VOCs to exactly that concentration where they start to show an effect. Until that concentration is reached the fungus proliferates first normally, then showing some phenotypic stress responses (change in color, less dense mycelium, etc) and eventually stopping to grow.
We included this explanation and discussion with published work in the new version of the manuscript in line 240-248. Additionally, we also would like to stress again, that we are working with clones. As the producing gene(s) are not in their natural background, but expressed recombinantly, it is likely that the produced VOCs are emitted at lower concentration. This is to say, that we don´t expect to get inhibition rates with these clones at the range equal to a really strong bacterial strain isolated from nature.
Finally, the pure compounds were tested at non-physiological concentrations. By physiological, I don’t mean ecologically relevant, but rather concentrations that are close to the experiment when a metagenomic bacterial clone inhibited fungal growth. If one is to use the experiment with pure compounds to prove that it is these molecules that cause inhibitory effects, the concentrations should be reasonable.
- Authors ´ response: We appreciate the clarification and the meaning of the raised concern is now clear to us. We carefully thought about the remark, but would like to refrain from quantifying the physiological concentration for the following reason:
Bacteria and fungi produce an ensemble of different VOCs, in some cases several dozen compounds and often more than just one VOCs shows an inhibitory effect. Thereby the inhibitory efficacy can differ, so that one compound might cause the observed inhibition to a higher share then other, also inhibitory VOCs. This means, even if the physiological concentration of the VOC of interest released into the air by the clone is quantified, it cannot be said how much this concentration contributes to the inhibition. As per that, it might even be misleading as readers could assume that the quantified concentration caused the described inhibitory effect to a 100%.
Determining the share of inhibition by the VOCs of interest would go beyond the constraints of our study and might even not be possible, because as soon as an unknown VOC is detected nothing can be said about its inhibitory interplay with the other VOCs and vice versa.
We understand the concern about the concentrations used when the pure compounds were tested. Of course, at first, these concentrations seem high, but it is important to consider, that the compounds were applied as liquids and not all of that liquid evaporates into the air. So, the effective volatile concentration may differ from the applied concentration of the pure, liquid compound. We elaborated on the raised concern and calculated the theoretically maximal concentration in the gas-phase of the petri plate (under the assumption of full evaporation during the course of the cultivation) (Table S1 and lines 169-178). For this theoretical calculation the applied concentration of the substances in the gas-phase (where the inhibition takes place) is in the µM to mM range. We share the reviewer´s opinion, that a better understanding about the concentrations is relevant. However, the quantification of VOCs under the employed assay conditions (petri dish, prolongated cultivation times) is technically very challenging and, in our opinion, lies out of the scope of the study. Our hypothesis was that it is possible to develop a high throughput assay for screening of antimicrobial volatiles and with the presented results we give proof-of-principle for the developed screening strategy (è from metagenomic DNA to identification of antifungal clones è antifungal substances è possible genes or pathways).
I would argue that at high enough concentrations, almost everything becomes toxic. With all this combined, I’m not sure that the results presented in this manuscript are publishable and not artefacts.
- Authors ´ response: Sure, as the saying goes, the dose makes the poison. However, fungi are very robust, very resilient organisms often intrinsically resistant to many antimicrobials (as known from the problems with antibiotic resistances). So naturally, they often require a high dose of a natural compound to be effectively inhibited. Besides that, the presented concentrations are those of the pure compounds in their liquid form and as stated above, the volatile concentration may likely be lower. In fact, the maximal theoretical concentrations in the gas-phase as calculated (Table S1) are in the µM to mM range.
Lastly, we would like to emphasize, that the key intention and aim of this paper is not to prove specific inhibitory efficacies of the identified VOCs, but to present a new method to identify antifungal VOCs. The highlight of this method is, that it allows to identify VOCs from not-cultivable microorganisms. This is important as most of the biologically active compounds cannot be prospected and exploited due to the great plate count anomaly.
REFERENCES
Gao, H. et al. (2018) ‘Research on volatile organic compounds from Bacillus subtilis CF-3: Biocontrol effects on fruit fungal pathogens and dynamic changes during fermentation’, Frontiers in Microbiology. doi: 10.3389/fmicb.2018.00456.
Raza, W. et al. (2015) ‘Production of volatile organic compounds by an antagonistic strain Paenibacillus polymyxa WR-2 in the presence of root exudates and organic fertilizer and their antifungal activity against Fusarium oxysporum f. sp. niveum’, Biological Control. doi: 10.1016/j.biocontrol.2014.09.004.
Yuan, J. et al. (2012) ‘Antifungal activity of bacillus amyloliquefaciens NJN-6 volatile compounds against Fusarium oxysporum f. sp. cubense’, Applied and Environmental Microbiology. doi: 10.1128/AEM.01357-12.